# Analysis of the utilization of traditional medicine in Korea over 10 years (2013–2022): A repeated cross-sectional study using national health insurance data

**Minjung Park**[1] ☯, **Yoon Jae Lee**[2] ☯, **Seungwon Shin** ®[3]*

**1** College of Korean Medicine, Gachon University, Seongnam, Republic of Korea, **2** Jaseng Spine and Joint Research Institute, Jaseng Medical Foundation, Seoul, Republic of Korea, **3** College of Korean Medicine, Sangji University, Wonju, Republic of Korea

☯ These authors contributed equally to this work.
* ssw.kmd@gmail.com

## Abstract

### Background

This study aims to figure out the changes in traditional Korean medicine (TKM) utilization over the last decade, using the complete dataset of all claims submitted to the National Health Insurance (NHI) system in Korea.

### Methods

This is a repeated cross-sectional study between 2013 and 2022. To observe TKM healthcare resources, we collected the number of TKM institutions, beds, and doctors. To observe changes in the utilization of TKM services, we collected the number of patients, claims, and medical expenses submitted to NHI. Data was analyzed according to inpatient/outpatient and hospitals/clinics. The compound annual growth rate was calculated.

### Results

The growth was observed in the number of TKM hospitals, beds, and doctors. The changes in the claim count and the number of patients told us that TKM is going through a structural shift into hospital-centered services, even though the total number of patients utilizing TKM services decreased. The increase in TKM medical expenses was not merely a result of general inflation but reflects an increase in utilization of TKM services.

### Conclusions

The key findings include a significant increase in inpatient services of TKM hospitals and medical expenses. The diversification of diagnostic and treatment methods highlights the evolving nature of TKM. Future research should focus on a more comprehensive study to examine the sociocultural and clinical factors influencing TKM utilization.

**Data availability statement:** All relevant data are within the manuscript and its Supporting Information files.

**Funding:** This research was supported by a grant of the Korea Health Technology R&D Project through the Korea Health Industry Development Institute (KHIDI), funded by the Ministry of Health & Welfare, Republic of Korea (grant number: RS-2024-00441486). The funders had no role in study design, data collection and analysis, decision to publish, or preparation of the manuscript.

**Competing interests:** The authors have declared that no competing interests exist.

## Introduction

The healthcare system in the Republic of Korea is bifurcated into conventional Western medicine (WM) and traditional Korean medicine (TKM), like mainland China or Taiwan [1,2]. This dual system is sustained with distinct educational pathways, licensing requirements, and medical institutions dedicated to each type of medicine [3,4]. Such a structure allows for specialized training and practice in both domains, ensuring comprehensive healthcare coverage [5,6]. Korea's National Health Insurance (NHI) system, initiated in 1977, has achieved coverage for approximately 97–98% of the entire population since 1995 [6]. The NHI has also covered diagnostic or therapeutic services of TKM since 1985, suggesting that the Korean government recognizes the unique contributions of TKM to public health.

There has been a growing interest in understanding the trends and patterns related to utilizing traditional, complementary, and alternative medicine worldwide. The World Health Organization published a global report, in 2019, to monitor progress in traditional medicine over the past two decades, mostly focused on regulations and policies [7]. A national survey showed that the use and expenditures on alternative medicine in the United States increased significantly between 1990 and 1997 [8]. Another nationwide survey conducted in China revealed that 19.3% of Chinese adults aged 45 and older used traditional Chinese medicine, which was primarily driven by the presence of the chronic conditions [9]. It was also reported, from the analysis of the NHI database, that approximately one-fourth of beneficiaries in Korea (25.5%) and Taiwan (26.8%) used traditional medicine services in 2011 [10]. Another recent study presented that 25.4% of Koreans used TKM annually between 2008 and 2017 and medical expenditures of TKM accounted for around 4% of NHI in Korea [11].

This study aims to analyze the changes in TKM utilization over the last decade (2013–2022), using the complete dataset of all claims submitted to the NHI system operated by the Health Insurance Review & Assessment Service (HIRA) in Korea. Specifically, we focus on several key indicators for medical resources (TKM institutions, beds, and doctors) and medical utilization (patients, claims, and the medical expenses) to provide an evidence-based trend of TKM utilization. By examining these indicators, we expect to offer a comprehensive overview of how TKM utilization has evolved over the last 10 years.

## Meterials and methods

This is a repeated cross-sectional study collecting data at multiple time points from the entire population, allowing researchers to observe changes and trends over time without following the same individuals. This differs from a longitudinal study tracking the same individuals over time, providing insights into individual-level changes and developments [12].

### Sources of data

We used the Bigdata Open Portal Data database run by HIRA Service in Korea, which provides access to comprehensive health insurance claims data, allowing anyone to download and use statistics on resources and utilization of Korean medical services for research and education purposes [13]. We extracted, summarized, and analyzed annual aggregate statistics on indicators of interest over ten years between 2013 and 2022 (the most recent year when data is available as of June 8 2024) from the database. The authors did not have access to information that could identify individual participants during or after data collection. Given that the NHI coverage rate in Korea is approximately 97–98% [6], this research should be considered a census, not a sample survey.

### Indicators of interest

To observe TKM healthcare resources, we selected the indicators of the annual number of medical institutions, beds, and medical doctors, which are separately summarized according to types of TKM institutions (TKM hospitals primarily for inpatients and TKM clinics primarily for outpatients). The number of beds is known to serve as a critical indicator for estimating healthcare service capacity [14]. The number of medical institutions and beds in TKM was also compared annually with those of the entire medical resource in Korea. We also examined the number of WM doctors in TKM hospitals and TKM doctors in WM hospitals to observe changes in cooperative healthcare service patterns. In Korea, it is possible to establish WM departments in TKM hospitals, or vice versa. If necessary, the data of the same indicators for WM hospitals and clinics, which are leveled similar to TKM hospitals and clinics, respectively in the Korean healthcare system's hierarchy, were also compared.

To observe changes in the utilization of TKM healthcare services, we collected the number of patients & health insurance claims and medical expenses over ten years. All values are analyzed separately for inpatient/outpatient and TKM hospitals/clinics, as well as their aggregated totals. The medical expenses per patient or claim were calculated. Additionally, the annual rate of change and its average rate, i.e., the compound annual growth rate (CAGR) were calculated as follows:

$$Yearly\ Change\ Rate\ (\%) = \frac{(value\ of\ the\ next\ year) - (value\ of\ the\ previous\ year)}{(value\ of\ the\ previous\ year)} \times 100$$

$$CAGR\ (\%) = \left( \left( \frac{value\ of\ the\ ending\ year}{value\ of\ the\ beginning\ year} \right)^{1/(number\ of\ years)} - 1 \right) \times 100$$

Furthermore, we tried to overview the changes in TKM service patterns by analyzing the relative composition ratios of the respective diagnostic and therapeutic methods covered by NHI in Korea, in terms of the number of claims and their medical expenses. The diagnostic methods were categorized into meridian function test, pulse wave examination, Yangdorak test, and dizziness or psychiatric examinations, while the therapeutic methods were into manual acupuncture, electroacupuncture, cupping therapy, moxibustion, Chuna manual therapy, and others.

### Data analysis and visualization

All data were analyzed using OpenAI's ChatGPT 4.0 and R program 4.3.2 (R Core Team. R: A Language and Environment for Statistical Computing. R Foundation for Statistical Computing) To visually convey the characteristics of the data effectively, this paper presents changes in each indicator by bar and line plots rather than tables. Detailed data values are provided in the supporting information. Since this study aimed to identify trends in TKM utilization rather than test specific hypotheses, significance testing was not performed.

### Ethics approval and consent to participate

All procedures were performed in compliance with relevant laws and institutional guidelines. Since the study did not involve the direct recruitment of human subjects or the analysis of individual participant data, the ethical review was exempted, including informed consent by the Institutional Review Board of Sangji University (Wonju, Republic of Korea) on June 7, 2024 (Approval No. 2024-08).

## Results

### Changes in healthcare resources: TKM institutions, beds, and doctors

The aggregated proportion of TKM hospitals and clinics out of total medical institutions in Korea has remained relatively stable, averaging around 15.55% between 2013 and 2022. However, the number of TKM hospitals increased significantly from 212 to 546, with a CAGR of 11.08%, while TKM clinics grew from 13,100–14,549, with a CAGR of 1.17%. Considering that 1) the CAGR of all the medical institutions in Korea over 10 years was 1.87%, 2) the number of WM hospitals decreased from 1,415–1,398, and 3) the number of WM clinics showed a growth rate at a CAGR of 2.36%, it is notable that the number of TKM hospitals significantly increased and that of TKM clinics stayed mostly unchanged (Figs 1(a) and (1b)).

The number of beds in TKM hospitals increased from 12,819–34,378, with a CAGR of 11.58%. Similarly, TKM clinic beds grew from 3,155–7,933, showing a CAGR of 10.79%. In contrast, the total number of beds across all institutions in Korea increased from 630,114–724,212, with a CAGR of 1.56%. During the same period, the number of beds in WM hospital and clinic decreased with CAGRs of -4.16 and -5.60%, respectively (Figs 1 (c) and (1d)).

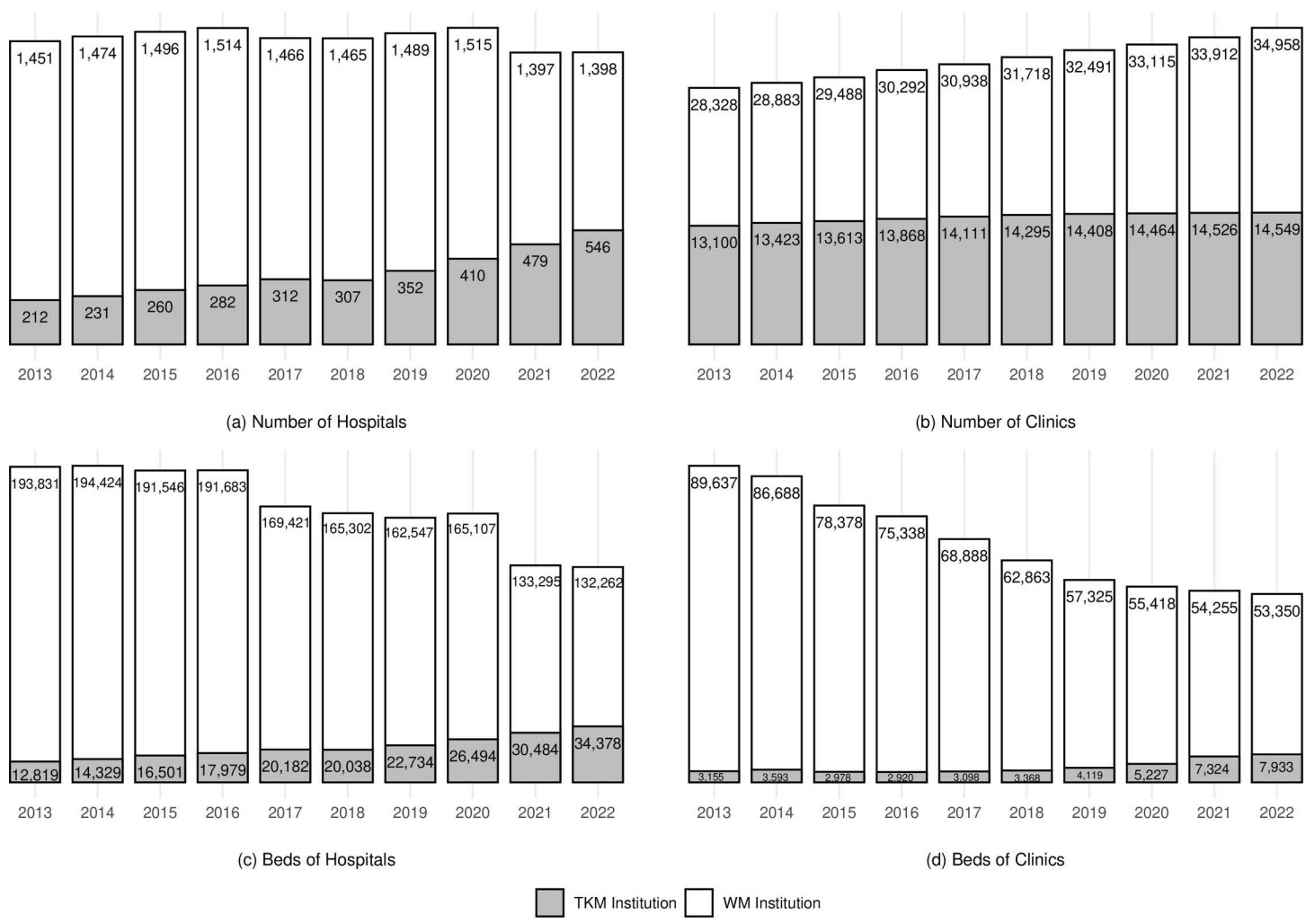

**Fig 1. Medical institutions and beds in Korea between 2013 and 2022.** TKM: Traditional Korean Medicine, WM: Western Medicine.

The total number of TKM doctors increased from 18,199–22,807 with a CAGR of 2.54%, which is similar to the change of WM doctors during the same period. However, the number of TKM doctors in hospitals grew from 18,199–22,807, showing a CAGR of 2.54%. The most interesting change appeared in the number of WM doctors working in TKM hospitals, which increased substantially from 91 to 559, with a CAGR of 22.35% (Fig 2).

The annual values for each indicator are provided in the S1 Table, which presents the number of medical institutions, beds, and medical doctors in Korea between 2013 and 2022.

## Changes in medical service utilization: patients, claims, and medical expenses

The total number of patients who submitted claims to HIRA for TKM services experienced a decrease from 13,346 thousand (2013) to 11,585 thousand (2022), with a CAGR of -1.56%, which contrasts with the number of patients in WM services remaining relatively stable over the same ten-year period (CAGR of 0.78%). It was shown that the patients had

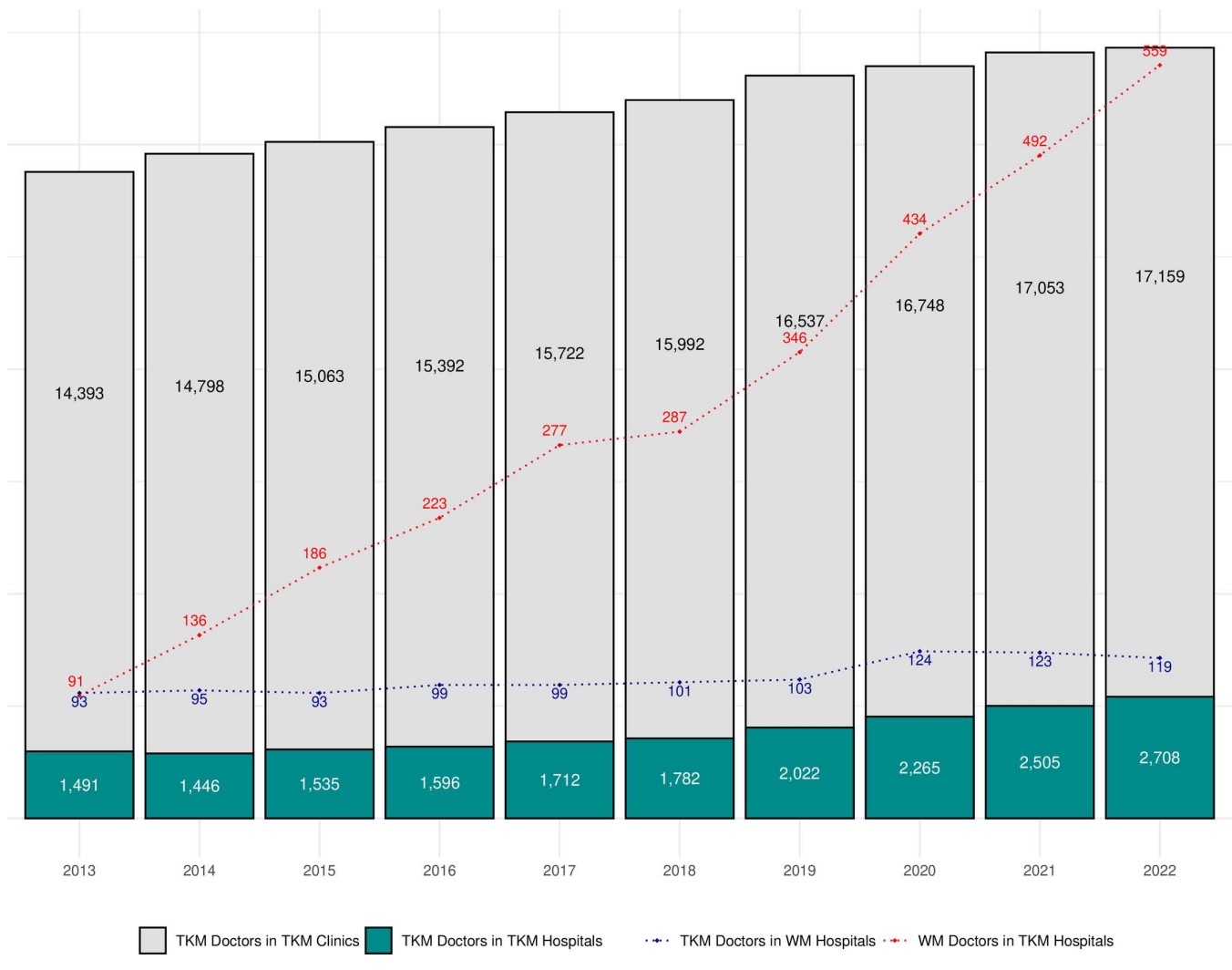

**Fig 2. Traditional medical doctors in Korea between 2013 and 2022.** TKM: Traditional Korean Medicine, WM: Western Medicine.

negative annual growth since 2015. However, there is a notable divergence when differentiating between inpatients and outpatients. The number of inpatients in TKM institutions increased from 114 thousand (2013) to 177 thousand (2022), with a CAGR of 5.01%, while the number of outpatients showed a CAGR of -1.62% (Fig 3 (a1)). This discrepancy becomes more pronounced when examining the patient numbers separately for TKM hospitals and clinics. The number of patients in TKM increased from 689 thousand to 849 thousand over 10 years (inpatients with a CAGR of 5.30% and outpatients with a CAGR of 1.73%) (Fig 3 (a2)), whereas the number of patients in TKM clinics decreased with a CAGR of -1.81% (Fig 3 (a3)).

The number of claims showed a trend similar to the change in patient numbers. Total claims in TKM institutions decreased from 101,126 thousand in 2013–91,302 thousand in 2022, with a CAGR of -1.13%. However, when separating inpatient and outpatient claims, inpatient claims increased significantly with a CAGR of 9.57%, while outpatient claims decreased with a CAGR of -1.17% (Fig 3 (b1)). Analysis of claims by institution type revealed that claims for inpatient services at TKM hospitals increased by approximately 9.99%

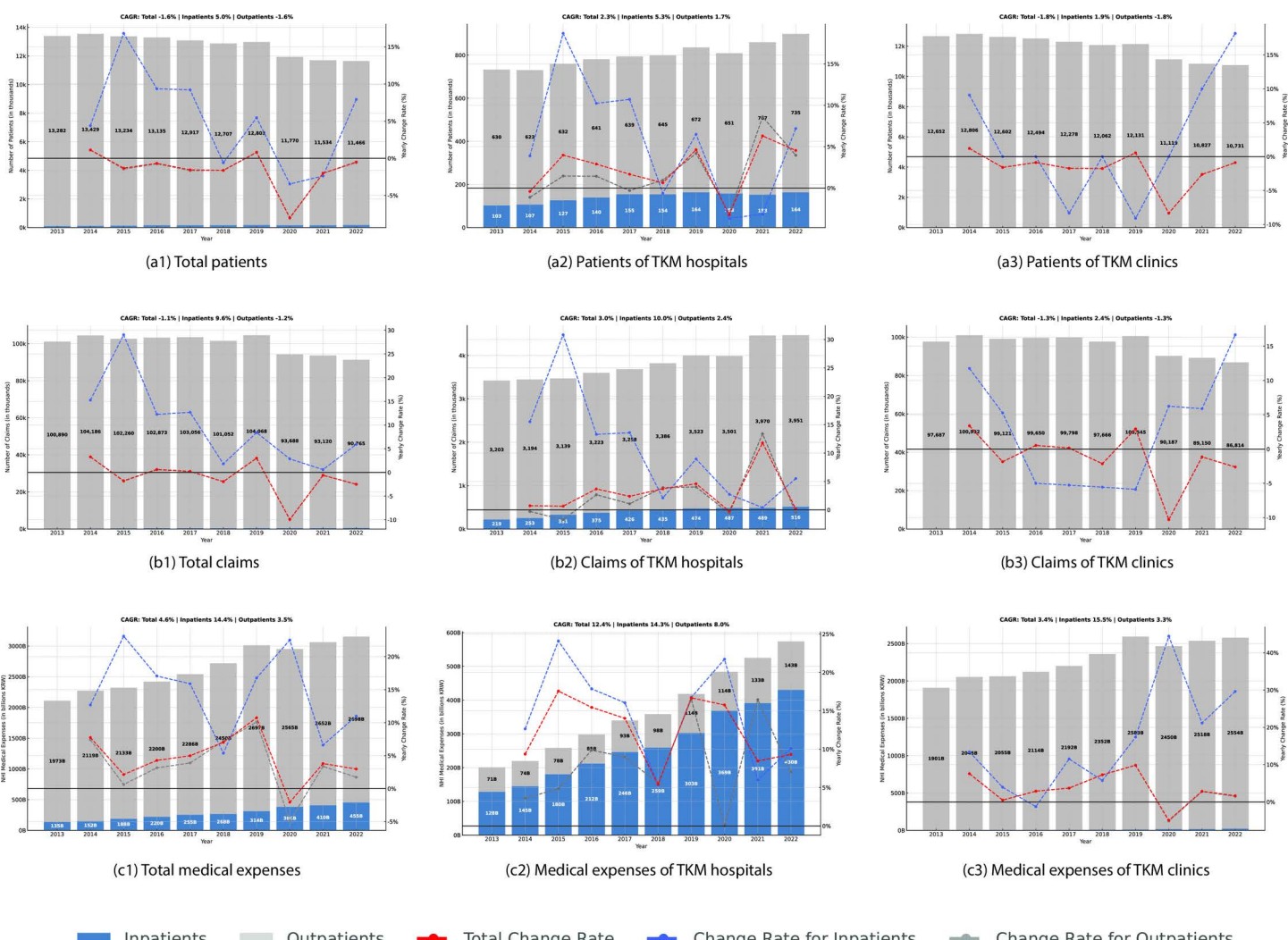

**Fig 3. Patients, claims, and medical expenses for TKM services between 2013 and 2022.** CAGR: Compound Annual Growth Rate, TKM: Traditional Korean Medicine, WM: Western Medicine.

(Fig 3 (b2)), and outpatient claims increased by 2.36%, contrasting with the decline in claims at TKM clinics (Fig 3 (b3)).

The total medical expenses reimbursed to TKM institutions increased from 2,109 billion Korean won (KRW) in 2013–3,153 billion KRW in 2022, with a CAGR of 4.57%. This growth appears to be driven by increased medical expenses associated with inpatient care (CAGR of 14.40%). Excluding 2020, the year the COVID-19 pandemic began, there has been a positive annual growth rate every year since 2013 (Fig 3 (a3)). The significant increase in inpatient expenses is more pronounced when looking into the data of TKM hospitals, separately. In TKM hospitals, inpatient expenses rose from 128 to 430 billion KRW, with a CAGR of 14.34%, while outpatient expenses increased from 71 to 143 billion KRW, with a CAGR of 7.97% (Fig 3 (c2)). In contrast, inpatient expenses in WM hospitals grew at a CAGR of 5.25% and outpatient expenses grew at a CAGR of 9.41% during the same period, highlighting a distinguishing trend in expenses change between TKM and WM institutions.

The medical expenses per patient in TKM institutions increased from 158,000–272,000 KRW. Notably, the expenditures per inpatient rose significantly from 1,189,000–2,572,000 KRW, with a CAGR of 8.95%. The expenses per claim also increased from 20,855–34,526 KRW with a CAGR of 5.76%, which is lower than the annual growth rate of 7.35% observed in WM institutions.

It is noteworthy that all indicators, including the number of patients, claims, and medical expenses, showed a serious decline during the coronavirus disease 2019 (COVID-19) pandemic (2020–2021). Still, there was a substantial increase in claims and medical expenses for inpatient care at TKM hospitals and clinics during the same period despite the overall decrease.

The specific data for each indicator are provided in the S2 to S4 Tables, which present the number of patients (S2 Table), the number of claims (S3 Table), and medical expenses reimbursed by national health insurance (S4 Table) in Korea between 2013 and 2022.

## Changes in medical service utilization: distribution of medical examination or treatments used in TKM

Meridian function tests consistently held the highest proportion of claim counts and medical expenses over 10 years, followed by pulse wave examination, Yangdorak test, personality examination, dizziness examination, and dementia examination. Meridian function tests dominated both inpatient and outpatient services. Yangdorak test showed a decreasing trend in both claims (5.69% in 2013 to 3.39% in 2022) and expenses (3.26% in 2013 to 3.07% in 2022), whereas pulse wave examination (8.27% in 2013 to 16.79% in 2022 for claims, 10.92% in 2013 to 14.93% in 2022 for expenses, respectively) and personality examination (0.21% in 2013 to 1.26% in 2022 for claims, 0.78% in 2013 to 5.02% in 2022 for expenses, respectively) exhibited growth. Dizziness and dementia tests remained minimal throughout the period (Figs 4 (a) and 4(b)).

Manual acupuncture had the highest proportion of claim counts and medical costs, followed by cupping therapy. Manual acupuncture dominated both claim counts and costs, but its composition ratio decreased over the ten years (59.48% in 2013 to 50.05% in 2022 for claims, 63.05% in 2013 to 47.72% in 2022 for expenses, respectively). In contrast, cupping therapy showed increases in claims (12.76% in 2013 to 15.09% in 2022) and costs (16.93% in 2013 to 21.65% in 2022) over the ten-year period. Electroacupuncture and moxibustion therapies also demonstrated a substantial increase in usage. The most notable change was observed in Chuna manual therapy. The treatment began to show claims in 2019 and revealed a steep increase of 51.8% in claim counts and of 64.4% in cost for 3 years (2019–2022) (Figs 4 (c) and 4(d)).

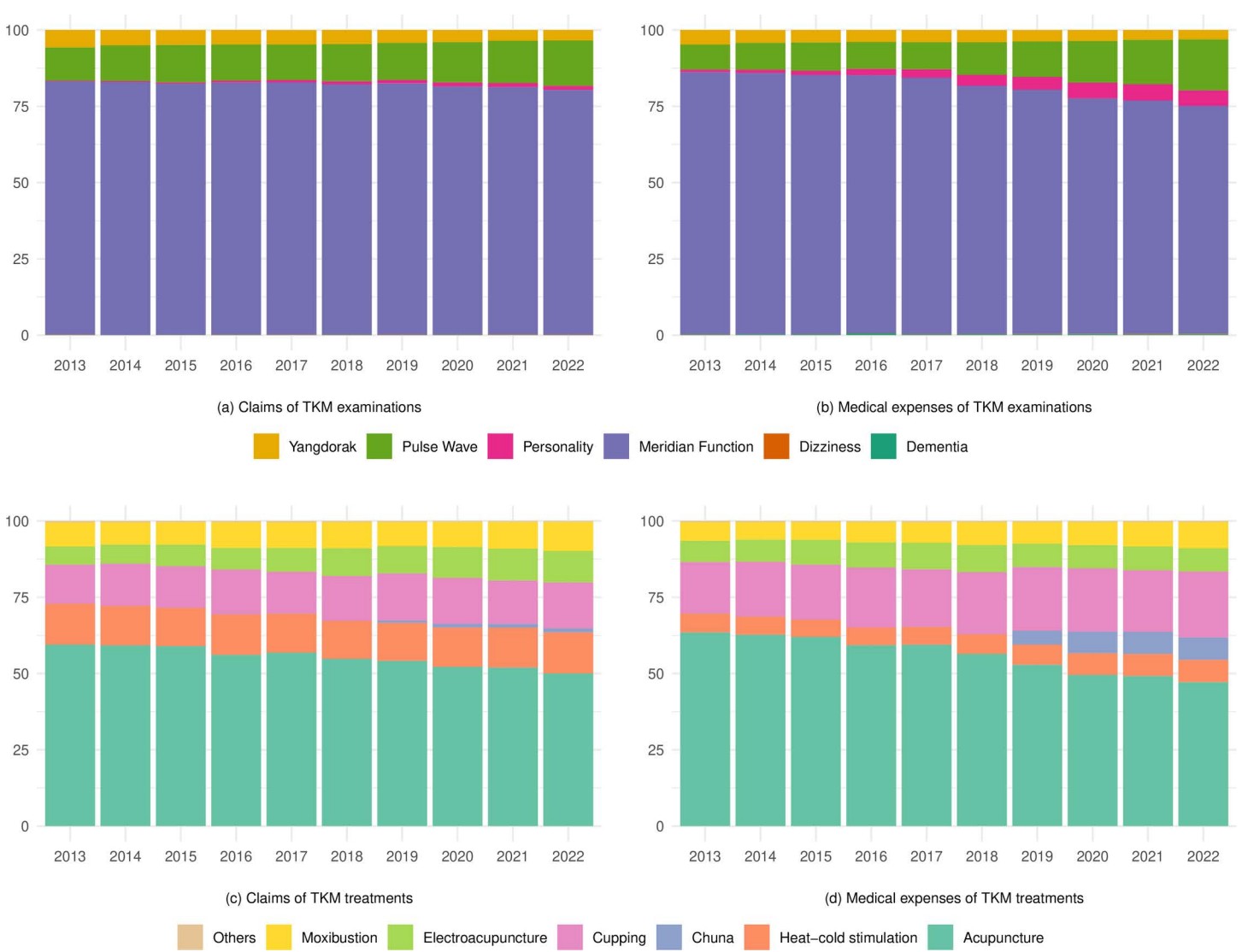

**Fig 4. Distribution of medical examinations and treatments of TKM between 2013 and 2022.** The unit of the Y-axis is %.

The specific data for each indicator are provided in the S5 and S6 Tables, which detail the claims and medical expenses for TKM examinations (S5.1 and S5.2 Tables) and treatments (S6.1 and S6.2 Tables) in Korea between 2013 and 2022.

## Discussion

### Key findings

This was a repeated cross-sectional study using comprehensive health insurance claims data from HIRA in Korea, aiming to analyze the changes in TKM utilization over the last decade (2013–2022).

Significant growth was observed in the number of TKM hospitals (CAGR of 11.08%) and beds (CAGR of 11.58%), while the number of TKM clinics remained relatively stable. The total number of TKM doctors increased (CAGR of 2.54%), but the number of WM doctors working in TKM hospitals rose more substantially (CAGR of 22.35%) over the decade. The

total number of patients utilizing TKM services decreased (CAGR of -1.56%), primarily due to a decline in outpatient. However, inpatient claims (CAGR of 9.57%) and medical expenses (CAGR of 14.40%) saw significant increases. Total medical expenses for TKM institutions also increased, driven by the rise in inpatient care costs. About the changes in the composition of TKM diagnostic methods, meridian function tests consistently held the highest proportion of claims and medical expenses, while pulse wave and personality examinations showed growth over the decade. Among treatment methods, manual acupuncture maintained the highest proportion but decreased over time, whereas cupping therapy, electroacupuncture, and moxibustion therapies demonstrated increases in usage. Notably, Chuna manual therapy exhibited a steep increase in claim counts (51.8%) and in costs (64.4%) since 2019.

## Implication of the results

First, the observed decline in the number of patients and claims for TKM service over the past decade can be interpreted in terms of the changes in the overall population of Korea. The official population increased slightly from 50,429 thousand in 2013–51,673 thousand in 2022 (CAGR of 0.98%) [15], resulting in a decrease in the proportion of TKM users relative to the total population from 26.5% to 22.4%. This seems relatively more decrease, compared to the previous studies on TKM utilization conducted in other countries [16] such as Taiwan (26.8% in 2011) [10], the United States (42.1% in 1997) [8], and Denmark (24% in 2021) [17], and the CAGR of patients in WM institutions (0.78%) during the same period in Korea. The significant slowdown in population growth may have had a more substantial impact on the utilization of TKM.

One of the possible factors contributing to the reduction in TKM utilization in Korea may have been the impact of the COVID-19 pandemic, considering that the number of patients and claims experienced severe negative growth between 2020 and 2021(Fig 3). COVID-19 reduced access to healthcare services and altered health-seeking behaviors due to lockdowns, social distancing measures, and fear of infection in healthcare settings. Previous studies reported that the impact of COVID-19 on complementary and alternative medicine was considerable [18,19]. Another study also reported that Korea experienced a reduction of medical utilization in outpatient by 15.7% and inpatient by 11.6% in overall healthcare service [20]. A quantitative analysis such as an interrupted time-series analysis is needed to determine whether COVID-19 has influenced the changes in the number of TKM users in the future.

Second, the increase in medical expenses reimbursed to TKM services over the research period (CAGR of 4.57%), particularly driven by inpatient care, reflects an increasing reliance on TKM for more intensive treatments within the Korean healthcare system. The medical expenses per patient or claim also increased, suggesting payers are willing to spend more on TKM services. Even though the average rates of change are lower than those of WM institutions, they outpace general inflation in Korea. The Consumer Price Index (CPI) of Korea rose from 93.0 in 2013 to 107.7 in 2022 (2020 as the base year with CPI of 100) at a CAGR of approximately 1.57% [15]. This disparity indicates that the increase in TKM medical expenses is not merely a result of inflation but reflects a real increase in the cost and utilization of TKM services.

Third, this study indicates the growing importance of hospitals within the TKM system. Over the past decade, there has been a notable increase in healthcare resources for TKM hospitals, which is mirrored by the rise in the number of patients and claims in TKM hospitals. Several factors may contribute to this trend. As revealed in previous studies, patients prefer hospitals over clinics due to the perceived higher quality of care, advanced medical technology, and better-equipped facilities, as well as trust and the reputation of hospitals [21–23].

Another contributing factor may be the increasing interest in integrative or collaborative medicine in Korea. Although the Korean medical system is bifurcated into WM and TKM in terms of education, licensing, diagnosis, and treatment [3,4], there is a growing need for cooperation between the two medical fields. This is facilitated, for example, by establishing WM departments within TKM hospitals (not clinics), allowing for integrative or collaborative care. Public preference for collaborative care might have resulted in the growth of TKM hospitals, which is supported by one of the findings that the number of WM doctors working in TKM hospitals increased (Fig 2).

Lastly, there has been a diversification in the diagnostic and treatment methods provided by TKM over the past decade. Even though meridian function tests consistently held the highest proportion of claim counts and medical expenses, the expansion of personality examinations highlights the growing inclusion of psychiatric diagnostic tests in TKM. The treatment methods in TKM have also become more varied. While manual acupuncture is holding the highest proportion of claim counts and medical costs, other treatments such as cupping therapy, electroacupuncture, and Chuna manual therapy have shown significant growth. These changes are largely influenced by the evolving health insurance coverage for TKM. The inclusion of new diagnostic and treatment methods, such as Chuna manual therapy, in the NHI scheme has played a crucial role in their adoption.

The role of traditional medicine within national healthcare systems varies considerably across the globe. In the United States, complementary and alternative medicine (CAM) operates largely outside the conventional insurance framework, with utilization driven by consumer demand and out-of-pocket spending [24]. Mainland China has integrated traditional Chinese medicine (TCM) into its national healthcare system with government support [25]. Taiwan's national health insurance system covers TCM services, contributing to its increased utilization in that population [26]. In contrast, the United Kingdom's National Health Service (NHS) offers limited coverage for traditional medicine practices such as acupuncture, often relying on private insurance or out-of-pocket payments [27]. Korea's TKM system, with its national health insurance coverage and distinct category of licensed doctors, represents a unique model that balances traditional practices with modern healthcare infrastructure.

In line with the World Health Organization's strategies promoting traditional and complementary medicine [7], there is growing global recognition of its potential role in healthcare. In the United States, CAM has shown steady growth, with acupuncture being a popular modality [24]. Similarly, Taiwan has seen increasing use of TCM among older adults [26]. In mainland China, the TCM healthcare workforce has been greatly strengthened, especially in primary healthcare settings [25], reflecting the government's emphasis on strengthening TCM services. Our study highlights the growing importance of TKM hospitals in Korea and the increasing integration of TKM with WM, suggesting a potential convergence of healthcare approaches. Further research is needed to understand the long-term implications of these changes on healthcare delivery.

## Strengths and limitations of the study

This study possesses several strengths. First, the study analyzed comprehensive data from authorized health insurance claims. Considering that the NHI coverage in Korea is approximately 97–98% [6], the results of this study are highly representative of the whole population, allowing it to be generalized to understand trends in TKM utilization. Second, this research encompasses data up to 2022, the most recent data available as of April 2024, compared with the previous study conducted with the data from 2008 to 2017 [11]. This can provide the latest trends and perspectives of TKM, including the impacts of the COVID-19 pandemic period. Third, this study goes beyond merely explaining the trends in individual indicators of TKM utilization. It comprehensively examines the structural changes within TKM, such as the

increasing importance of TKM hospitals and the diversification of medical services provided. As a result, this study could provide a multi-faceted view of how TKM is evolving to meet the healthcare needs of the population.

Despite the strengths of this study, several limitations must be acknowledged. First, this study did not include all TKM institutions in Korea. Public health centers and care hospitals providing TKM services were excluded due to data availability constraints. Second, the study did not cover TKM services provided through other insurance schemes such as medical aid or private automobile insurance. Third, this study did not examine the factors influencing TKM usage over the ten years. Previous research indicates that factors such as gender, age, income, and specific diseases can affect preferences for TKM [3,27–31]. This could be explored through a longitudinal study, like cohort or case-control study, in the future. Also, since our study employed a repeated cross-sectional design, we could not track individual-level changes in TKM utilization over time. While we observed trends at the population level, we were unable to determine how individual TKM users' behaviors changed during the study period. Furthermore, the repeated cross-sectional design makes it difficult to establish causal relationships between potential risk factors and TKM utilization at the individual level. Fourth, the claims data may not fully reflect the actual clinical practice. For example, medical services not covered by insurance are excluded, potentially underestimating TKM utilization. Selective billing by medical institutions may also lead to over- or under-representation of specific diagnoses or treatments. Finally, we could not obtain data on herbal drugs, which are a crucial treatment method in TKM. This limitation is particularly significant since herbal medicine constitutes a substantial portion of TKM practice. The exclusion of herbal medicine data may result in an underestimation of TKM utilization. However, the Korean government has recently implemented a pilot project to expand insurance coverage for herbal drugs, which may enable future studies to include herbal medicine data to evaluate overall TKM utilization. A more comprehensive understanding of TKM utilization in the future can be achieved, potentially leading to better healthcare strategies and policies.

## Conclusion

This study has provided a comprehensive analysis of the trends and changes in the utilization of TKM from 2013 to 2022. Key findings include a significant increase in inpatient services of TKM hospitals and medical expenses. The diversification of diagnostic and treatment methods highlights the evolving nature of TKM. Despite the study's limitations, the findings offer valuable insights into the structural and functional transformations within TKM. Future research should focus on a more comprehensive study including all TKM-related data and examine the sociocultural and clinical factors influencing TKM utilization. Understanding these patterns in the evidence-based framework is essential for policymakers, healthcare providers, and researchers to promote the effective integration of TKM within the broader healthcare system, thereby improving public health outcomes in Korea.

## Supporting information

**S1 Table. Number of medical institutions, beds, and medical doctors in Korea between 2013 and 2022.**
(PDF)

**S2 Table. Number of patients in Korea between 2013 and 2022.**
(PDF)

**S3 Table. Number of claims in Korea between 2013 and 2022.**
(PDF)

**S4 Table. Medical expenses reimbursed by national health insurance in Korea between 2013 and 2022.**
(PDF)

**S5.1 Table. Claims for TKM examinations in Korea between 2013 and 2022.**
(PDF)

**S5.2 Table. Medical expenses for TKM examinations in Korea between 2013 and 2022.**
(PDF)

**S6.1 Table. Claims for TKM treatments in Korea between 2013 and 2022.**
(PDF)

**S6.2 Table. Medical expenses for TKM treatments in Korea between 2013 and 2022.**
(PDF)

## Acknowledgments

This study utilized medical statistical information from 2013 to 2022 provided by the Health Insurance Review & Assessment Service (HIRA) Big Data Open Portal (opendata.hira.or.kr) in the Republic of Korea.

## Author contributions

**Conceptualization:** Seungwon Shin.

**Data curation:** Seungwon Shin.

**Methodology:** Minjung Park, Yoon Jae Lee, Seungwon Shin.

**Project administration:** Seungwon Shin.

**Visualization:** Seungwon Shin.

**Writing – original draft:** Minjung Park, Yoon Jae Lee, Seungwon Shin.

**Writing – review & editing:** Seungwon Shin.

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
