## [Decision Letter · Decision Letter 0]

17 Feb 2025

PONE-D-24-34363Analysis of the utilization of traditional medicine in Korea over 10 years (2013-2022): A repeated cross-sectional study using national health insurance dataPLOS ONE

Dear Dr. Shin,

Thank you for submitting your manuscript to PLOS ONE. After careful consideration, we feel that it has merit but does not fully meet PLOS ONE’s publication criteria as it currently stands. Therefore, we invite you to submit a revised version of the manuscript that addresses the points raised during the review process.

Please focus on detailed comments from reviewer 1 and revise your manuscript adequately.

We look forward to receiving your revised manuscript.

Kind regards,

Marcelo Dionisio

Academic Editor

PLOS ONE

Journal Requirements:

“This research was supported by a grant of the Korea Health Technology R&D Project through the Korea Health Industry Development Institute (KHIDI), funded by the Ministry of Health & Welfare, Republic of Korea (grant number: RS-2024-00441486). This study utilized medical statistical information from 2013 to 2022 provided by the Health Insurance Review & Assessment Service (HIRA) Big Data Open Portal (opendata.hira.or.kr) in the Republic of Korea. ”

“This research was supported by a grant of the Korea Health Technology R&D Project through the Korea Health Industry Development Institute (KHIDI), funded by the Ministry of Health & Welfare, Republic of Korea (grant number: RS-2024-00441486).”

“This research was supported by a grant of the Korea Health Technology R&D Project through the Korea Health Industry Development Institute (KHIDI), funded by the Ministry of Health & Welfare, Republic of Korea (grant number: RS-2024-00441486).”

Reviewers' comments:

Reviewer's Responses to Questions

**Comments to the Author**

1. Is the manuscript technically sound, and do the data support the conclusions?

Reviewer #1: Yes

Reviewer #2: Yes

2. Has the statistical analysis been performed appropriately and rigorously? 

Reviewer #1: Yes

Reviewer #2: Yes

3. Have the authors made all data underlying the findings in their manuscript fully available?

Reviewer #1: Yes

Reviewer #2: Yes

4. Is the manuscript presented in an intelligible fashion and written in standard English?

Reviewer #1: Yes

Reviewer #2: Yes

5. Review Comments to the Author

Reviewer #1: Expand the global relevance of TKM by referencing WHO policies or comparisons with other countries using traditional medicine systems.

The manuscript does not explain potential biases in claims data.

The description of statistical methods, such as CAGR calculation, is brief and lacks details on significance testing.

The repeated cross-sectional design limitations (e.g., inability to track individuals) are not adequately discussed.

The results acknowledge a decline during COVID-19 but do not provide sufficient detail.

Minimal discussion of how trends in Korea compare to other countries with traditional medicine systems.

Herbal drugs, an essential component of TKM, are excluded but this limitation is underemphasized.

Supporting data tables (S1–S6) are referenced but not adequately summarized in the main text.

Reviewer #2: The title

• It is clear, informative, and representative of the content and breadth of the study

The abstract:

• It is complete and essential details are presented

The introduction/background

• It gives exhaustive literature information and describes the knowledge gap but lacks a research question for the intended study.

The research design

• It is appropriate and clearly defined, sample size looks reasonable.

Data analysis:

• It is sufficiently described and sufficiently detailed to permit the study to be replicated.

Results:

• It is well organized in a way that is easy to understand

Tables, graphs, or figures

• They are used judiciously and agree with the text.

The conclusions and recommendations

• They are clearly stated; key points stand out.

Recommendation as reviewer

• The manuscript adheres to the journal guidelines and standards.

• I fully agree with the acceptance of the manuscript after making minor revisions, particularly the necessary editorials.

6. PLOS authors have the option to publish the peer review history of their article (what does this mean? ). If published, this will include your full peer review and any attached files.

**Do you want your identity to be public for this peer review?** For information about this choice, including consent withdrawal, please see our Privacy Policy .

Reviewer #1: No

Reviewer #2: No

---

## [Author Response · Author response to Decision Letter 1]

19 Feb 2025

- For the editor’s comments:

→ We appreciate the manuals for manuscript preparation. We checked the PDF files and revised the manuscript format as needed.

“This research was supported by a grant of the Korea Health Technology R&D Project through the Korea Health Industry Development Institute (KHIDI), funded by the Ministry of Health & Welfare, Republic of Korea (grant number: RS-2024-00441486). This study utilized medical statistical information from 2013 to 2022 provided by the Health Insurance Review & Assessment Service (HIRA) Big Data Open Portal (opendata.hira.or.kr) in the Republic of Korea. ”

“This research was supported by a grant of the Korea Health Technology R&D Project through the Korea Health Industry Development Institute (KHIDI), funded by the Ministry of Health & Welfare, Republic of Korea (grant number: RS-2024-00441486).”

→ We appreciate the comments. We removed the funding-related text from the manuscript. The amendment of the funding statement is as follows:

This research was supported by a grant of the Korea Health Technology R&D Project through the Korea Health Industry Development Institute (KHIDI), funded by the Ministry of Health & Welfare, Republic of Korea (grant number: RS-2024-00441486). The funders had no role in study design, data collection and analysis, decision to publish, or preparation of the manuscript.

“This research was supported by a grant of the Korea Health Technology R&D Project through the Korea Health Industry Development Institute (KHIDI), funded by the Ministry of Health & Welfare, Republic of Korea (grant number: RS-2024-00441486).”

→ We appreciate the comments. The amendment of the funding statement is as follows:

This research was supported by a grant of the Korea Health Technology R&D Project through the Korea Health Industry Development Institute (KHIDI), funded by the Ministry of Health & Welfare, Republic of Korea (grant number: RS-2024-00441486). The funders had no role in study design, data collection and analysis, decision to publish, or preparation of the manuscript.

→ We appreciate the comments. We moved the ethics statement to the Methods section.

→ We have carefully reviewed our reference list and verified all citations against the original sources to ensure they are accurate and complete. We have also added references to support new content added to the revised manuscript. We confirm that none of the cited papers have been retracted.

- For the reviewer 1’s comments:

1. Expand the global relevance of TKM by referencing WHO policies or comparisons with other countries using traditional medicine systems.

→ We appreciate the reviewer's insightful comment regarding the need to expand the global relevance of TKM. We have carefully considered this feedback and have added the following paragraph in the discussion section to address this point more comprehensively.

In line with the World Health Organization's strategies promoting traditional and complementary medicine, (7) there is growing global recognition of its potential role in healthcare. In the United States, CAM has shown steady growth, with acupuncture being a popular modality. (24) Similarly, Taiwan has seen increasing use of TCM among older adults. (26) In mainland China, the TCM healthcare workforce has been greatly strengthened, especially in primary healthcare settings, (25) reflecting the government's emphasis on strengthening TCM services. Our study highlights the growing importance of TKM hospitals in Korea and the increasing integration of TKM with WM, suggesting a potential convergence of healthcare approaches. Further research is needed to understand the long-term implications of these changes on healthcare delivery.

7. World Health Organization. WHO global report on traditional and complementary medicine 2019. World Health Organization; 2019.

24. Jin LL, Zheng J, Honarvar NM, Chen X. Traditional Chinese Medicine in the United States: Current state, regulations, challenges, and the way forward. Traditional Medicine and Modern Medicine. 2020 Mar 13;03(02):77–84.

25. Liu Y, Yang L, Du C, Schernhammer ES, Giovannucci EL, Hou PC, et al. The traditional Chinese medicine health-care workforce in mainland China: 10-year trend analysis of nationwide data. The Lancet. 2019;394:S38.

26. Huang CJ, Chang CC, Chen TL, Yeh CC, Lin JG, Liu CH, et al. The long-term trend in utilization of traditional Chinese medicine and associated factors among older people in Taiwan. PLoS One. 2024;19(5).

2. The manuscript does not explain potential biases in claims data.

→ We appreciate the reviewer's valuable comment regarding the potential biases in claims data. We agree that this is an important limitation and have added the following sentences to the limitations section:

Fourth, the claims data may not fully reflect the actual clinical practice. For example, medical services not covered by insurance are excluded, potentially underestimating TKM utilization. Selective billing by medical institutions may also lead to over- or under-representation of specific diagnoses or treatments.

The description of statistical methods, such as CAGR calculation, is brief and lacks details on significance testing.

→ We appreciate the reviewer's comment regarding the description of statistical methods. We have already presented the CAGR calculation formula in the Methods section. We would also like to clarify that this study aimed to describe trends in TKM utilization over time, meaning that it was not designed to test specific hypotheses requiring significance testing. The following sentence is added in the Method section to clarify this.

3. Since this study aimed to identify trends in TKM utilization rather than test specific hypotheses, significance testing was not performed.

The repeated cross-sectional design limitations (e.g., inability to track individuals) are not adequately discussed.

→ We appreciate the reviewer's comment regarding the limitations of our repeated cross-sectional design. We agree that this is an important limitation and have added the following sentences to the limitations section:

Also, since our study employed a repeated cross-sectional design, we could not track individual-level changes in TKM utilization over time. While we observed trends at the population level, we were unable to determine how individual TKM users' behaviors changed during the study period. Furthermore, the repeated cross-sectional design makes it difficult to establish causal relationships between potential risk factors and TKM utilization at the individual level.

4. The results acknowledge a decline during COVID-19 but do not provide sufficient detail.

→ We appreciate the reviewer's comment regarding the need for a more detailed analysis of the impact of COVID-19 on TKM utilization. Our manuscript indicated a notable decline in TKM utilization during the COVID-19 pandemic (2020-2021). However, due to the limitations of our repeated cross-sectional design, we were unable to conduct a more in-depth analysis of the specific factors driving this decline. To address this limitation, we acknowledge the need for future studies employing time-series analysis to assess the impact of COVID-19 on TKM utilization quantitatively. For example, a recent study by Lee et al. (2023) used interrupted time-series analysis to examine the impact of the COVID-19 pandemic on overall healthcare utilization in South Korea. We believe that future research employing similar methodologies is warranted to gain a more comprehensive understanding of the impact of COVID-19 on TKM utilization.

(Yoo KJ, Lee Y, Lee S, Friebel R, Shin SA, Lee T, Bishai D. The road to recovery: impact of COVID-19 on healthcare utilization in South Korea in 2016-2022 using an interrupted time-series analysis. Lancet Reg Health West Pac. 2023 21;41:100904.)

5. Minimal discussion of how trends in Korea compare to other countries with traditional medicine systems.

→ We appreciate the reviewer's comment regarding the need for a more detailed discussion of how trends in Korea compare to other countries with traditional medicine systems. We agree that providing a comparative analysis can enhance the global relevance of our findings. The following paragraph is added to the Discussion section.

The role of traditional medicine within national healthcare systems varies considerably across the globe. In the United States, complementary and alternative medicine (CAM) operates largely outside the conventional insurance framework, with utilization driven by consumer demand and out-of-pocket spending. (24) Mainland China has integrated traditional Chinese medicine (TCM) into its national healthcare system with government support. (25) Taiwan's national health insurance system covers TCM services, contributing to its increased utilization in that population. (26) In contrast, the United Kingdom's National Health Service (NHS) offers limited coverage for traditional medicine practices such as acupuncture, often relying on private insurance or out-of-pocket payments. (27) Korea's TKM system, with its national health insurance coverage and distinct category of licensed doctors, represents a unique model that balances traditional practices with modern healthcare infrastructure.

24. Jin LL, Zheng J, Honarvar NM, Chen X. Traditional Chinese Medicine in the United States: Current state, regulations, challenges, and the way forward. Traditional Medicine and Modern Medicine. 2020;03(02):77–84.

25. Liu Y, Yang L, Du C, Schernhammer ES, Giovannucci EL, Hou PC, et al. The traditional Chinese medicine health-care workforce in mainland China: 10-year trend analysis of nationwide data. The Lancet. 2019;394:S38.

26. Huang CJ, Chang CC, Chen TL, Yeh CC, Lin JG, Liu CH, et al. The long-term trend in utilization of traditional Chinese medicine and associated factors among older people in Taiwan. PLoS One. 2024;19(5).

27. Hopton AK, Curnoe S, Kanaan M, MacPherson H. Acupuncture in practice: mapping the providers, the patients and the settings in a national cross-sectional survey. BMJ Open. 2012;2(1):e000456.

6. Herbal drugs, an essential component of TKM, are excluded but this limitation is underemphasized.

→ We appreciate the reviewer's comment regarding the exclusion of herbal drug data and its potential impact on the study's findings. We agree this is a significant limitation, so we have revised the following sentences.

Finally, we could not use data on herbal drugs, which is an important treatment method of TKM. Since most herbal drugs are still prescribed as non-covered items in Korea, we could not acquire the relevant data from the HIRA database.

↓

Finally, we could not obtain data on herbal drugs, which are a crucial treatment method in TKM. This limitation is particularly significant since herbal medicine constitutes a substantial portion of TKM practice. The exclusion of herbal medicine data may result in an underestimation of TKM utilization. However, the Korean government has recently implemented a pilot project to expand insurance coverage for herbal drugs, which may enable future studies to include herbal medicine data to evaluate overall TKM utilization.

7. Supporting data tables (S1–S6) are referenced but not adequately summarized in the main text.

→ We appreciate the reviewer's comment regarding the need for a more detailed summary of the supporting data tables in the main text. In response, we have added sentences to each relevant section of the Results section to indicate the specific data presented in Tables S1-S6 clearly.

The annual values for each indicator are provided in the S1 Table, which presents the number of medical institutions, beds, and medical doctors in Korea between 2013 and 2022.

The specific data for each indicator are provided in the S2 to S4 Tables, which present the number of patients (S2 Table), the number of claims (S3 Table), and medical expenses reimbursed by national health insurance (S4 Table) in Korea between 2013 and 2022.

The specific data for each indicator are provided in the S5 and S6 Tables, which detail the claims and medical expenses for TKM examinations (S5.1 and S5.2 Tables) and treatments (S6.1 and S6.2 Tables) in Korea between 2013 and 2022.

For the reviewer 2’s comments:

The title

• It is clear, informative, and representative of the content and breadth of the study

The abstract:

• It is complete and essential details are presented

The introduction/background

• It gives exhaustive literature information and describes the knowledge gap but lacks a research question for the intended study.

The research design

• It is appropriate and clearly defined, sample size looks reasonable.

Data analysis:

• It is sufficiently described and sufficiently detailed to permit the study to be replicated.

Results:

• It is well organized in a way that is easy to understand

Tables, graphs, or figures

• They are used judiciously and agree with the text.

The conclusions and recommendations

• They are clearly stated; key points stand out.

Recommendation as reviewer

• The manuscript adheres to the journal guidelines and standards.

• I fully agree with the acceptance of the manuscript after making minor revisions, particularly the necessary editorials.

→ We appreciate all the reviewer’s comments.

Other revisions:

One of the first authors (Minjung Park) has changed her affiliation due to a job change.

→ 1 College of Korean Medicine, Gachon University, Seongnam, Republic of Korea

---

## [Decision Letter · Decision Letter 1]

7 Mar 2025

Analysis of the utilization of traditional medicine in Korea over 10 years (2013-2022): A repeated cross-sectional study using national health insurance data

PONE-D-24-34363R1

Dear Dr. Shin,

We’re pleased to inform you that your manuscript has been judged scientifically suitable for publication and will be formally accepted for publication once it meets all outstanding technical requirements.

Kind regards,

Marcelo Dionisio

Support Staff - Editorial

PLOS ONE

Additional Editor Comments (optional):

Reviewers' comments:

Reviewer's Responses to Questions

**Comments to the Author**

1. If the authors have adequately addressed your comments raised in a previous round of review and you feel that this manuscript is now acceptable for publication, you may indicate that here to bypass the “Comments to the Author” section, enter your conflict of interest statement in the “Confidential to Editor” section, and submit your "Accept" recommendation.

Reviewer #3: All comments have been addressed

Reviewer #4: All comments have been addressed

2. Is the manuscript technically sound, and do the data support the conclusions?

Reviewer #3: Yes

Reviewer #4: Yes

3. Has the statistical analysis been performed appropriately and rigorously? 

Reviewer #3: Yes

Reviewer #4: Yes

4. Have the authors made all data underlying the findings in their manuscript fully available?

Reviewer #3: Yes

Reviewer #4: Yes

5. Is the manuscript presented in an intelligible fashion and written in standard English?

Reviewer #3: Yes

Reviewer #4: Yes

6. Review Comments to the Author

Reviewer #3: Thank you for your valuable article. It is valuable to examine the status of traditional medicine in order to analyze its benefits and disadvantages.

Reviewer #4: I am pleased to recommend the acceptance of this manuscript for publication in your esteemed journal. The study provides a comprehensive analysis of the trends and changes in the utilization of Traditional Korean Medicine (TKM) from 2013 to 2022, offering valuable insights into the structural and functional transformations within TKM.

The key findings, including the significant increase in inpatient services of TKM hospitals and medical expenses, as well as the diversification of diagnostic and treatment methods, highlight the evolving nature of TKM. These observations are crucial for understanding the current landscape of TKM and its integration into the broader healthcare system.

While the study acknowledges some limitations, its contributions to the field are substantial. The call for future research to include all TKM-related data and to examine sociocultural and clinical factors influencing TKM utilization is well-justified. Such comprehensive studies are essential for policymakers, healthcare providers, and researchers aiming to promote the effective integration of TKM and improve public health outcomes in Korea.

Overall, the manuscript is well-structured, and the conclusions are supported by the data presented. I believe that this study will contribute significantly to the ongoing discussions about the role of traditional medicine in modern healthcare systems.

Thank you for the opportunity to review this manuscript.

7. PLOS authors have the option to publish the peer review history of their article (what does this mean? ). If published, this will include your full peer review and any attached files.

**Do you want your identity to be public for this peer review?** For information about this choice, including consent withdrawal, please see our Privacy Policy .

Reviewer #3: No

Reviewer #4: No

---

## [Editor Report · Acceptance letter]

PONE-D-24-34363R1

PLOS ONE

Dear Dr. Shin,

I'm pleased to inform you that your manuscript has been deemed suitable for publication in PLOS ONE. Congratulations! Your manuscript is now being handed over to our production team.

Kind regards,

on behalf of

Dr. PLOS Manuscript Reassignment

Staff Editor

PLOS ONE